# Effect of Uncertainty in Illness and Fatigue on Health-Related Quality of Life of Patients on Dialysis: A Cross-Sectional Correlation Study

**DOI:** 10.3390/healthcare10102043

**Published:** 2022-10-16

**Authors:** Ok-Hee Cho, Insook Hong, Hyekyung Kim

**Affiliations:** 1Department of Nursing, College of Nursing and Health, Kongju National University, Gongju 32588, Korea; 2Department of Nursing, Gyeongbuk College of Health, Gimcheon 39525, Korea; 3Department of Nursing, Catholic Kwandong University, Gangneung 25601, Korea

**Keywords:** dialysis, fatigue, quality of life, uncertainty

## Abstract

This study aimed to determine the effect of uncertainty in illness and fatigue on the health-related quality of life of patients on dialysis. A community-based cross-sectional study was conducted among patients on hemodialysis (*n* = 80) and peritoneal dialysis (*n* = 81) in Korea. Data were collated using self-reported structured questionnaires. Multiple regression analysis was used to identify those factors affecting the physical and mental health-related quality of life of patients. Patients on peritoneal dialysis reported higher levels of fatigue (*p* < 0.001). Factors affecting the physical health-related quality of life of patients on dialysis were fatigue (*p* < 0.001), employment (*p* = 0.001), and exercise (*p* = 0.016), thus explaining the observed variance of 37%. Factors affecting mental health-related quality of life were fatigue (*p* < 0.001), uncertainty (*p* = 0.004), educational level (*p* = 0.005), and smoking (*p* = 0.035). To improve the health-related quality of life of patients on dialysis, clinicians should assess their fatigue levels and plan multidisciplinary interventions to manage it. In addition, education level and employment status should be considered, and tailored interventions should be provided to acquire positive coping strategies and health promotion behaviors to counter disease uncertainty.

## 1. Introduction

Dialysis is one of the main treatment methods replacing renal function in patients. In Korea, the number of patients on dialysis has nearly doubled, from 69,986 in 2015 to 123,122 in 2020 [1]; between 2009–2010 and 2017–2018, the prevalence of patients on dialysis increased by 75%—the third-highest increase after Indonesia and Thailand [2]. With the development of dialysis treatment technology, the management of patients’ conditions has been prolonged, and the goal of treatment is not to eliminate the disease but to adapt to the patient’s physical limitations, lifestyle changes, medical treatment, and quality of life (QoL) as important performance indicators [3]. Generally, the health-related quality of life (HRQoL) of patients on dialysis is worse than that of people of the same age in the general population because of the high burden of comorbidities and complications inherent in chronic kidney disease (CKD) [4]. In a large prospective study [5], approximately two-thirds (68%) of patients on dialysis reported a decrease in QoL, which represented more than 60% of CKD patients and 49% of kidney transplant (KT) patients. A meta-analysis of papers published over the course of 30 years also reported that patients on dialysis had a lower QoL than patients after KT [6]. Sociodemographic characteristics, such as age, educational level and economic status [7], gender and employment [8], disease and treatment-related factors, such as the presence of comorbidities or complications [5], dialysis modality [6], dialysis period [8], health behavioral-related factors, such as smoking [7], exercise [9], diet [10], fatigue [11], uncertainty [12], and anxiety, depression, and family support [5] affected HRQoL in patients on dialysis.

Patients on dialysis live their entire lives with the threat of death if treatment is stopped and suffer uncertainty about unforeseen circumstances and complications [12]. When patients perceive uncertainty as a threat of psychological stress, the probability of wrong decision-making increases [13], and their QoL can be impaired, with reduced adherence to treatment and lesser coping ability [14]. Therefore, understanding the relationship between the patients’ perceived uncertainty and QoL is an important topic for providing insight into the structure of their QoL.

Fatigue is one of the most common symptoms complained of by more than 70% of patients on dialysis [11]; it is known that the cause of this fatigue is related to demographic factors, physiological conditions, psychological conditions, biochemical factors, and behavioral factors [15,16]. Patients have the highest level of fatigue immediately after dialysis but may complain of fatigue symptoms, such as weakness, lack of energy, and poor concentration, even on those days without dialysis [17]. These factors cause patients to experience fatigue and low QoL. Current CKD guidelines recommend that patients actively monitor their symptoms, quit smoking, maintain a healthy weight, eat healthily, and exercise to manage the disease and alleviate its sequelae [18]. Patients with low adherence to these recommended health behaviors are associated with an increased risk of poor clinical outcomes [19]. Therefore, identifying health-related behaviors that are associated with the QoL of patients on dialysis will facilitate the screening of high-risk groups and help provide effective tailored interventions. However, depending on the dialysis modality (hemodialysis or peritoneal dialysis), the dialysis site, self-management requirements, necessary equipment, supplies and drugs, and dialysis-related symptoms are different. Therefore, the dialysis modality must be considered to understand the daily life and QoL of patients on dialysis [6,20]. Despite the recent surge in the population on dialysis, there are still very few studies on the QoL of patients on peritoneal dialysis and hemodialysis in Korea. Therefore, this study was conducted, first, to identify differences in the QoL according to dialysis modality and, second, to provide basic data for developing interventions by identifying factors affecting the QoL of patients on dialysis.

## 2. Materials and Methods

### 2.1. Study Design

This study bore a cross-sectional, correlational, and descriptive design and used questionnaires to identify those factors affecting the QoL of patients on dialysis.

### 2.2. Participants and Data Collection

The study participants were recruited using the snowball method; they included the inhabitants of four Korean regions who were ≥19 years old and were receiving hemodialysis at a primary medical institution or peritoneal dialysis at home. Patients with acute renal failure or kidney transplantation were excluded. The sample size was calculated using the G*power 3.1.9.4 software (Heinrich-Heine-Universität Düsseldorf, Düsseldorf, Germany). With a statistical significance of 0.05, a power of 0.8, a medium effect size of 0.15, and predictors in multiple regression, the minimum sample size necessary was found to be 135. Assuming a 20% dropout rate, 161 patients were enrolled in this study.

Sociodemographic characteristics included age, gender, marital status, the possibility of self-care, religion, subjective economic status, education, employment, time period of dialysis, dialysis modality (hemodialysis/peritoneal dialysis), and health-related behaviors, including smoking, drinking, and exercise. The data collection period was from October 2019 to January 2020. As for the data collection method, the purpose of this study and questionnaire were explained in small informal groups of 2–3 people, gathered through the introduction of patients undergoing hemodialysis or peritoneal dialysis, and a face-to-face survey was conducted directly with the participants. It took about 15–20 min to complete the questionnaire, and a predetermined gift was provided to the participants after completing the questionnaire.

### 2.3. Instruments

#### 2.3.1. Uncertainty in Illness

Uncertainty in illness was measured in the Korean version of the “Uncertainty in Illness Scale—Community Form” (MUIS-C) [21] developed by Mishel (1997) [22]. MUIS-C consists of 23 questions, categorized into four subscales: ambiguity, complexity, inconsistency, and unpredictability. The uncertainty score was the sum of these four subgroups and ranged between 23 and 115. The highest possible total score was 115, with higher scores indicating a greater level of uncertainty. Using a 5-point Likert-type scale, the responses ranged from “SA” (strongly agree) to “SD” (strongly disagree). Cronbach’s α was 0.81 for the present study.

#### 2.3.2. Fatigue

Fatigue was measured using the Korean version of the multidimensional assessment of fatigue (MAF) [23] developed by Tack [24]. The MAF assesses the degree of fatigue by employing 16 items, using a numeric rating scale. It measures four dimensions of fatigue: severity, distress, degree of interference in the activities of daily living, and timing. MAF scoring is used to calculate the global fatigue index, which ranges from 1 to 50. Higher scores indicate higher fatigue levels; Cronbach’s α was 0.95 for the present study.

#### 2.3.3. Health-Related QoL

Health-related QoL was measured in the Korean version of the short form-12 health survey questionnaire (SF-12) developed by Ware and Sherbourne [25]. The SF-12 is a shortened version of the Short Form-36 Health Survey Questionnaire and consists of 12 questions that measure eight health domains to assess the physical component summary score (PCS) and mental component summary score (MCS). The PCS score includes general health, physical functioning, physical role, and bodily pain. The MCS score includes vitality, social functioning, emotional role, and mental health. The scoring used in the tool was calculated based on the manual provided, and the data from the completed questionnaires were converted to 0–100 metrics. The higher the score, the better the QoL. In this study, Cronbach’s α was 0.85 (PCS = 0.74, MCS = 0.75).

### 2.4. Data Analysis

The collected data were analyzed using SPSS WIN 27.0 (IBM Corp., Armonk, NY, USA). The participants’ general characteristics, uncertainty, fatigue, and QoL were analyzed using descriptive statistics. Differences in QoL according to general characteristics were identified using Scheffé’s test. Correlations of continuous variables were identified using Pearson’s correlation coefficient, and factors affecting the QoL were identified using multiple regression analysis.

In order to check whether the assumption of regression analysis was justified, the correlation coefficient between independent variables (−0.43–0.17) was found to be less than 0.80, indicating independence between variables. The Durbin–Watson statistic (physical HRQoL 2.184, mental HRQoL 2.110) was close to 2, indicating the absence of autocorrelation. The tolerance limit (0.467–0.908) was > 0.1, and the dispersion expansion factor (1.101–2.141) was ≤10. As a result of residual analysis, linearity, normality, and equal variance were satisfied.

## 3. Results

### 3.1. General Characteristics of Participants

Of the total 161 participants, 80 were treated with peritoneal dialysis and 81 were treated with hemodialysis. The average age of participants was 54.5 years; 57.8% of participants were male. Most participants were married (67.1%), under self-care by themselves (85.7), of no particular religion (63.3%), were high school graduates (44.0%), and were unemployed (63.3%). The mean time period of dialysis was 7.63 years. Of the participants, 85.7% were non-smokers, 90.7% were non-drinkers, and 49.1% exercised irregularly (Table 1).

### 3.2. Differences in Uncertainty, Fatigue, and QoL According to Dialysis Modalities

The average fatigue level of all participants was 21.92 ± 10.35, and patients on peritoneal dialysis reported a higher level of fatigue (24.89 ± 10.49) than hemodialysis patients (18.94 ± 9.36) (t = −3.79, *p* < 0.001). The participants’ average HRQoL in the physical component summary score (PCS) was 36.57 ± 8.55 out of 50, and their average HRQoL of the mental component summary score (MCS) was 45.87 ± 16.56 (Table 2).

### 3.3. Factors Influencing Physical HRQoL

From the multiple regression analysis to identify factors affecting the physical HRQoL of patients on dialysis, among the statistically significant factors identified, fatigue (β = −0.34, *p* < 0.001) had the greatest influence, followed by employment (β = 0.26, *p* = 0.001) and exercise (β = 0.17, *p* = 0.016). These factors accounted for 37% of the variance of physical HRQoL (R^2^ = 0.37, adjusted R^2^ = 0.30, F = 5.57, *p* < 0.001) (Table 3).

### 3.4. Factors Influencing Mental HRQoL

The results of multiple regression analysis to identify the factors influencing the mental HRQoL of dialysis patients show that, among the statistically significant factors, fatigue (β = −0.48, *p* < 0.001) had the greatest influence, followed by uncertainty (β = −0.22, *p* = 0.004), educational level (β = −0.22, *p* = 0.005), and smoking (β = 0.15, *p* = 0.035). These factors accounted for 41% of the variance of physical HRQoL (R^2^ = 0.41, adjusted R^2^ = 0.35, F = 6.75, *p* < 0.001) (Table 4).

## 4. Discussion

This study was conducted to provide a theoretical framework for the QoL of patients on dialysis by investigating the level of HRQoL and its influencing factors, as well as the basic data for developing interventions. HRQoL encompasses numerous multidimensional concepts [26]. Identifying and understanding the key factors affecting the QoL across physical and mental domains is necessary when laying the groundwork for the implementation of targeted therapies and interventions to reduce the risk and increase the protective factors in vulnerable patient populations.

In this study, fatigue, as perceived by patients on dialysis, was the main factor negatively affecting both their physical and mental HRQoL. This result was consistent with the findings of previous studies [27,28]. Fatigue in patients on dialysis is caused by various complex sociodemographic, biological, and psychological factors [15], affecting the overall QoL [11]. A study of patients on hemodialysis (*n* = 134) [27] reported that all participants complained of fatigue, of whom 15% bore a high or very high fatigue level, both of which are associated with decreased QoL. In a cross-sectional cohort study [28], 53.3% of patients on hemodialysis complained of fatigue, which is higher than was reported by patients with kidney transplantation (33.3%), patients with hematological disorders in remission (23.3%), and the control group containing healthy participants (12.1%). Additionally, the fatigue of patients on dialysis was associated with a lower QoL than other patient groups. In this study, it was confirmed that the fatigue of patients on dialysis, which is implicitly taken for granted, is a problem that must be dealt with first, in order to improve the QoL. Even if the cause of fatigue in patients on dialysis is unclear, medical staff should continue improving the QoL by applying periodic fatigue evaluations [15] and fatigue relief interventions [16].

In this study, the uncertainty perceived by dialysis patients negatively affected only their mental HRQoL. This result is slightly different from a previous report stating that uncertainty negatively affected physical and mental HRQoL in a cross-sectional study [12] of patients on maintenance hemodialysis. However, Delis reported that uncertainty could negatively affect mental HRQoL in patients with chronic diseases, with a high incidence of mental problems such as depression [29]. In a study of patients with heart failure (*n* = 302) [30], it was reported that the uncertainty of the disease as perceived by the patient had a negative and indirect effect on mental HRQoL through perceived stress and acceptance/resignation. Research on prostate cancer patients (*n* = 263) [14] also reported that uncertainty had the indirect effect of lowering mental HRQoL through the avoidance of coping strategies, supporting the results of this study. An et al. and Guan et al. suggested that interventions that can help patients identify specific problems and reconstruct cognitive perceptions were effective in managing uncertainty [14,30]. In order to further clarify the relationship between uncertainty and QoL in dialysis patients in the future, it is necessary to explore the factors mediating these relationships. Because patients with chronic disease who respond positively to uncertainty will focus on managing their disease [31], providing related information, the management of complications, education on self-care [31], and the reinforcement of psychological support [14] may help to improve their mental HRQoL.

In this study, patients on dialysis who had a job had a higher physical HRQoL than patients without a job. A study of patients on dialysis [8] and patients with heart failure [30] supported this finding by reporting that having a job only positively affected their physical HRQoL. Since employment income provides the basis for more effective health management [32], it seems that it may have positively affected their physical HRQoL. However, Tannor et al. reported that patients on hemodialysis had to visit the hospital 2–3 times a week for dialysis, making it difficult to get a job, and even in the case of self-peritoneal dialysis, it was more difficult for them to find a job than for the general population, due to the time and place restrictions for dialysis fluid exchange [33]. In this study, only 36.7% of the participants were employed. Therefore, paying more attention to the physical needs of those patients with a job, helping them continue to work, and providing the necessary information and programs will improve their QoL. In the future, repeated studies are needed to include variables on specific job types, working hours, economic burden, and social support. It is also necessary to expand public services at the national level to provide support for patients who need a job for financial income [33].

It was confirmed that exercising (regularly or occasionally) positively affected the physical HRQoL of patients on dialysis. Previous studies found that exercise in patients on dialysis improved cardiovascular function, blood pressure, nutritional status, and dialysis quality, prevented muscle loss and bone disease [34], and was effective in reducing fatigue and improving sleep disorders [9]. Exercise in dialysis patients also positively improved their physical HRQoL by strengthening self-management activities [34]. In a meta-analysis study [9] on the effects of exercise on the symptoms of dialysis patients, it was reported that exercise was beneficial not only for the improvement of physical symptoms and function but also for mental health, such as depression and anxiety relief and mood improvement in dialysis patients. However, even though patients on dialysis were aware of the need for exercise, they were hesitant to exercise because of anemia due to dialysis, muscle fatigue, fear of arteriovenous fistula damage, or of falling during exercise [34]. Therefore, medical staff should plan a strategy to improve the QoL by evaluating physical activity and encouraging exercise in daily life, considering the health and environment of the patient undergoing dialysis.

In this study, high education status was a negatively influencing factor on mental HRQoL among patients on dialysis. This result was consistent with previous results demonstrating that the higher the educational level, the lower the mental HRQoL in patients with cancer [35]. Higher education can cause the patient to become more aware of the gap between the surrounding environment and the outside world, making it easier to be dissatisfied with the current situation [36]. Improving complications management and the ability to cope with unexpected situations in renal dialysis patients may have a positive effect on mental HRQoL [32]. Therefore, it would be helpful to suggest ways to improve positive coping skills while recognizing the current situation for those with a high level of education. However, several previous studies indicated that higher education levels may contribute positively to HRQoL [5,8]. Therefore, it is thought that confirmation through repeated studies is necessary.

Unexpectedly, the mental HRQoL of patients on dialysis who smoked was higher than that of non-smokers. A study investigating occupational stress and smoking in adults using data from the Health and Retirement Study (1992–2004) [37] found that smoking relieved stress. In this study, smoking may have been used as a habit developed by patients to control the stress caused by disease and the dialysis process. However, smoking is a major factor in worsening disease, increasing complications and mortality in patients on dialysis [19]; medical staff should be aware of the dangers of smoking, educate patients on dialysis to help them to quit smoking, and provide effective alternatives to smoking to reduce stress. The strategy for quitting smoking in dialysis patients can include behavioral therapy that strengthens the individuals’ motivations to quit smoking and pharmacotherapy [38]. In addition, it is also necessary to pay attention to the mental HRQoL of patients who do not smoke. The application of cognitive behavior therapy, excise, or relaxation techniques, presented as a conclusion of a meta-analysis on mental health interventions in dialysis patients, may be considered [39].

As described above, the findings that education, exercise, employment, and smoking were factors affecting the QoL indicate that individual characteristics and health behaviors can affect access to healthcare and the perception of QoL. This suggests that community-based social support, interventions, and a multidisciplinary approach are necessary for patients on dialysis to maintain their daily lives independently at home and improve their QoL.

## 5. Limitations

First, since this study recruited participants in certain Korean regions using convenience sampling, generalizing the study results should be performed cautiously. In particular, since the sampling was performed by voluntary consent, there is a possibility that participants with severely reduced QoL have been omitted. Second, being a cross-sectional study, this study obtained information at a single time point, so it was impossible to confirm the differences in QoL according to the course of the disease and the dialysis period. Rather than suggesting a causal relationship between QoL and related factors, only those correlations between variables were identified. Third, the disease and treatment characteristics and physiological indicators could not be controlled for. The factor that fatigue may be higher on the day of dialysis [17] was not considered. Fourth, recall bias could have occurred because subjectively perceived QoL and the level of related factors were identified. Fifth, the absence of a disease control group may be another limitation.

Nevertheless, this study analyzed the physical and mental factors affecting the QoL of patients on dialysis. Importantly, it provided evidence for selecting high-risk groups by checking the QoL and related factors controlling the dialysis modality. In the context of the recent surge in the number of patients on dialysis living in communities in Korea, investigating their QoL may assist in the balanced distribution of healthcare resources and community-based public health policies. A prospective, randomized, controlled, longitudinal study will investigate the causal relationship between QoL and related factors. In addition, we suggest conducting research related to changes in socioeconomic and health environment resources and support due to the pandemic, reviewing the impact of individual characteristics (resilience, self-efficacy, etc.) on the QoL, and the development and evaluation of quality-of-life interventions and their effectiveness.

## 6. Conclusions and Recommendations

In this study, self-reported fatigue, uncertainty, and personal characteristics (educational level, employment, exercise, and smoking) of patients on dialysis were confirmed as influencing factors on HRQoL. In particular, fatigue was the main factor to be considered for improving their physical and mental QoL. It may be helpful to provide periodic fatigue assessments and relief interventions to improve the patients’ QoL. In addition, it is necessary to pay attention to the physical HRQoL of unemployed, non-exercising patients, and the mental HRQoL of patients with high uncertainty, low education levels, and who do not smoke. These results can provide a useful basis for medical staff to plan effective support for dialysis patients to improve their QoL and tailor interventions to the specific disease.

## Figures and Tables

**Table 1 healthcare-10-02043-t001:** General characteristics of participants (*n* = 161).

Characteristics	Total	PD (*n* = 80)	HD (*n* = 81)
	*n* (%)	*n* (%)	*n* (%)
Sociodemographic			
Age, year	54.5 ± 13.5	48.8 ± 11.5	60.2 ± 13.1
Gender			
Male	93 (57.8)	42 (52.5)	51 (63.0)
Female	68 (42.2)	38 (47.5)	30 (37.0)
Marital status			
Unmarried	53 (16.8)	24 (18.8)	29 (14.8)
Married	108 (67.1)	56 (70.0)	52 (64.2)
Self-care			
Need assistance	23 (14.3)	5 (6.3)	18 (22.2)
By self	138 (85.7)	75 (93.7)	63 (77.8)
Religion			
Yes	59 (36.7)	26 (32.5)	33 (40.7)
No	102 (63.3)	54 (67.5)	48 (59.3)
Education			
Elementary school	16 (10.0)	5 (3.2)	11 (13.6)
Middle school	15 (9.3)	7 (8.8)	8 (9.9)
High school	71 (44.0)	34 (42.5)	37 (45.6)
College graduate	59 (38.7)	34 (42.5)	25 (30.9)
Subjective economic status			
High	12 (7.5)	6 (7.5)	6 (7.4)
Moderate	102 (63.3)	57 (75.3)	45 (55.6)
Poor	47 (29.2)	17 (21.2)	30 (37.0)
Employment			
Yes	59 (36.7)	34 (42.5)	25 (30.9)
No	102 (63.3)	46 (57.5)	56 (69.1)
Time period of dialysis (years)	7.63 ± 7.30	4.51 ± 3.71	10.70 ± 8.59
<1	15 (9.3)	8 (10.0)	7 (8.6)
1–5	62 (38.5)	46 (57.5)	16 (19.8)
>5	84 (52.2)	26 (32.5)	58 (71.6)
Health-related behaviors			
Smoking			
Yes	23(14.3)	7 (8.8)	16 (19.8)
No	138 (85.7)	73 (91.2)	65 (80.2)
Drinking			
Yes	15 (9.3)	7 (8.8)	8 (9.9)
No	146 (90.7)	73 (91.2)	73 (90.1)
Exercise			
Regular	27 (16.8)	30 (37.4)	25 (30.9)
Irregular	79 (49.1)	39 (48.8)	40 (49.3)
None	55 (34.1)	11 (13.8)	16 (19.8)

Note: HD = hemodialysis; PD = peritoneal dialysis.

**Table 2 healthcare-10-02043-t002:** Differences in uncertainty, fatigue, and health-related quality of life, according to the type of dialysis (*n* = 161).

Characteristics	Total	PD (*n* = 80)	HD (*n* = 81)	t	95% CI	*p*
	M ± SD	M ± SD	M ± SD			
Uncertainty in illness	60.01 ± 11.55	60.58 ± 11.23	59.46 ± 11.89	−0.61	−4.72–2.48	0.541
Fatigue	21.92 ± 10.35	24.89 ± 10.49	18.94 ± 9.36	−3.79	−9.06–−2.85	<0.001
Quality of life						
Physical functioning	36.57 ± 9.81	35.42 ± 9.36	37.70 ± 10.17	1.48	−0.77–5.32	0.142
Role (limitation) physical	36.95 ± 10.62	35.24 ± 9.92	38.64 ± 11.27	2.05	0.13–6.67	0.042
Bodily pain	44.78 ± 11.15	44.45 ± 11.35	45.11 ± 11.01	0.38	−2.82–4.15	0.707
General health	33.53 ± 8.82	32.88 ± 8.47	34.17 ± 9.16	0.93	−1.46–4.04	0.355
Vitality	44.87 ± 11.72	43.35 ± 11.30	46.38 ± 11.99	1.65	−0.59–6.66	0.100
Social functioning	41.39 ± 11.52	39.02 ± 11.10	43.73 ± 11.51	2.64	1.18–8.23	0.009
Role (limitation) emotional	38.71 ± 13.53	37.56 ± 12.79	39.86 ± 14.22	1.08	−1.91–6.51	0.282
Mental health	47.58 ± 9.92	46.41 ± 9.55	48.74 ± 10.19	1.50	−0.74–5.41	0.136
PCS-12	36.57 ± 8.55	35.67 ± 7.82	37.46 ± 9.18	1.33	−0.87–4.44	0.186
MCS-12	45.87 ± 16.56	44.32 ± 10.04	47.40 ± 10.89	1.87	−0.18–6.34	0.064

Note: HD = hemodialysis; PD = peritoneal dialysis; HRQoL = health-related quality of life; PCS = physical component summary score; MCS = mental component summary score; CI: confidence interval.

**Table 3 healthcare-10-02043-t003:** Factors affecting the physical health-related quality of life (*n* = 161).

Characteristics (Reference)	PCS-12				
	β	SE	Partial R^2^	95% CI	*p*
Intercept	0	6.71	0	33.95–60.47	<0.001
PD (vs. HD)	−0.12	1.54	0.011	−5.11–0.98	0.183
Sociodemographic					
Age, years	−0.03	0.06	0.049	−0.14–0.10	0.758
Male (vs. female)	0.04	1.47	0.063	−2.18–3.62	0.624
Married (vs. unmarried)	0.02	1.46	0.073	−2.51–3.26	0.799
High school/college graduate (vs. elementary/middle)	0.11	1.70	0.095	−1.09–5.64	0.184
Religious (vs. not religious)	−0.08	1.23	0.101	−3.92–0.95	0.231
Subjective economic status: high/moderate (vs. poor)	0.01	1.46	0.116	−2.72–3.04	0.912
Employment (vs. not employed)	0.26	1.37	0.161	1.93–7.33	0.001
Duration since dialysis, year	−0.12	0.09	0.171	−0.32–0.04	0.130
Self-care: need assistance (vs. by self)	−0.11	2.01	0.219	−6.72–1.24	0.176
Health-related behaviors					
Smoker (vs. not a smoker)	−0.04	1.81	0.219	−4.46–2.67	0.620
Drinker (vs. not a drinker)	−0.05	2.07	0.222	−5.67–2.50	0.443
Regular/irregular exercise (vs. none)	0.17	0.88	0.259	0.41–3.87	0.016
Uncertainty in illness	−0.09	0.06	0.287	−0.18–0.05	0.279
Fatigue	−0.34	0.08	0.367	−0.41–−0.15	<0.001

Note: HD = hemodialysis; PD = peritoneal dialysis; HRQoL = health-related quality of life; PCS = physical component summary score; CI confidence interval.

**Table 4 healthcare-10-02043-t004:** Factors affecting mental health-related quality of life (*n* = 161).

Characteristics (Reference)	MCS-12				
	β	SE	Partial R^2^	95% CI	*p*
Intercept	0	7.97	0	54.43–85.92	<0.001
PD (vs. HD)	0.019	1.83	0.020	−3.23–4.02	0.830
Sociodemographic					
Age, years	−0.04	0.07	0.021	−0.184–0.11	0.664
Male (vs. female)	0.04	1.74	0.025	−2.64–4.24	0.646
Married (vs. unmarried)	−0.07	1.73	0.026	−5.08–1.78	0.344
High school/college graduate (vs. elementary/middle)	−0.22	2.02	0.026	−9.77–1.78	0.005
Religious (vs. not religious)	0.03	1.46	0.032	−2.22–3.57	0.646
Subjective economic status: high/moderate (vs. poor)	−0.11	1.73	0.089	−6.00–0.84	0.138
Employment (vs. not employed)	0.03	1.62	0.090	−2.64–3.78	0.725
Duration since dialysis, year	0.06	0.11	0.092	−0.13–0.31	0.418
Self-care: need assistance (vs. by self)	0.07	2.39	0.095	−2.71–6.75	0.400
Health-related behaviors					
Smoker (vs. not a smoker)	0.15	2.14	0.126	0.33–8.80	0.035
Drinker (vs. not a drinker)	−0.08	2.45	0.137	−7.84–1.85	0.224
Regular/irregular exercise (vs. none)	0.07	1.04	0.149	−1.03–3.08	0.325
Uncertainty in illness	−0.22	0.07	0.256	−0.34–−0.07	0.004
Fatigue	−0.48	0.08	0.413	−0.64–−0.33	<0.001

Note: HD = hemodialysis; PD = peritoneal dialysis; HRQoL = health-related quality of life; MCS = mental component summary score; CI = confidence interval.

## Data Availability

Data are available upon request.

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
