# Peer review of "Effect of Uncertainty in Illness and Fatigue on Health-Related Quality of Life of Patients on Dialysis: A Cross-Sectional Correlation Study"

_healthcare, 2022, doi:10.3390/healthcare10102043_

Round 1

Reviewer 1 Report

This paper provides data useful for planning programs to advance physical and mental health among patients needing dialysis care. The quality of writing is fine. Although data were collected from a convenience sample, findings from the multivariable analysis can offer useful messages. 

There is one mistake that the authors need to revise. Data in Table 4 shows that patients with higher education attainment had lower mental health related quality of life, but the authors reported these patients had better mental health. Please confirm which direction was correct. If patients with higher education attainment had worse mental health, please elaborate more about how to help this type of patients.

Please also elaborate more about how to advance mental health among patients who were not smoking, and also propose possible intervention programs to help smoking patients to quit smoking without hurting their mental health.

Reviewer 2 Report

Nice work with very interesting results. Hemodialysis patients who are older and longer dialysis vintage on hemodialysis for longer have a better quality of life and fewer complaints. Undeniably, more investigations are to be conducted on a larger group of patients and in different populations.

In my opinion introduction  could be a bit shorter.

Reviewer 3 Report

Ok-Hee Cho, Insook Hong and Hye Kyung Kim investigated the health-related quality of life of the patients on dialysis by using structured questionnaires.  The study concluded fatigue, uncertainty, educational level, employment exercise  and smoking of the patients  could ininfluenced on HRQoL.  The study recommend that assessment of these factors are important for clinicians to improve the QoL of the patients on dialysis.

 According to Table 2, some factors except for General health and Mental health are significant difference between PD and HD. Although Uncertainty in illness, Bodily pain and Role in average are very small, this analysis showed significance.
1 I have a question regarding the results in this Table 2.What does the number behind the mean indicate, standard deviation or standard error?

2  Authors should indicate 95% confidence intervals for each item in Table 2.
